# Task-Oriented Feature Distillation

**Linfeng Zhang**[1*], **Yukang Shi**[2*], **Zuoqiang Shi**[1], **Kaisheng Ma**[1†], **Chenglong Bao**[1†]

Tsinghua University[1]     Xi'an Jiaotong University[2]

## Abstract

Feature distillation, a primary method in knowledge distillation, always leads to significant accuracy improvements. Most existing methods distill features in the teacher network through a manually designed transformation. In this paper, we propose a novel distillation method named task-oriented feature distillation (TOFD) where the transformation is convolutional layers that are trained in a data-driven manner by task loss. As a result, the task-oriented information in the features can be captured and distilled to students. Moreover, an orthogonal loss is applied to the feature resizing layer in TOFD to improve the performance of knowledge distillation. Experiments show that TOFD outperforms other distillation methods by a large margin on both image classification and 3D classification tasks. Codes have been released in Github[3].

## 1 Introduction

Recently, remarkable achievements have been attained with deep neural networks in all kinds of applications such as nature language processing [3, 13, 8, 60] and computer vision [54, 52, 51]. However, the success in neural networks is always accompanied by explosive growth of model parameters and computation, which has limited the deployment of neural networks on edge devices such as mobile phones and embedded devices. Various techniques have been proposed to tackle this issue, including pruning [15, 41, 72, 67, 39], quantization [35, 46, 11, 50], lightweight model design [23, 49, 27], and knowledge distillation (KD) [22, 53, 70].

Hinton *et al.* first propose the concept of distillation, where a lightweight student model is trained to mimic the softmax outputs (*i.e.* logit) of an over-parameterized teacher model [22]. Later, abundant feature distillation methods are proposed to encourage the student models to mimic the features of teacher models [53, 62, 66, 7, 20]. Since the features of teacher models have more information than logit, feature distillation enables student models to learn richer information and always leads to more accuracy improvements. As shown in Figure 1, instead of directly learning all the features of the teacher models, most of the feature distillation methods first apply a transformation function to the features to convert them

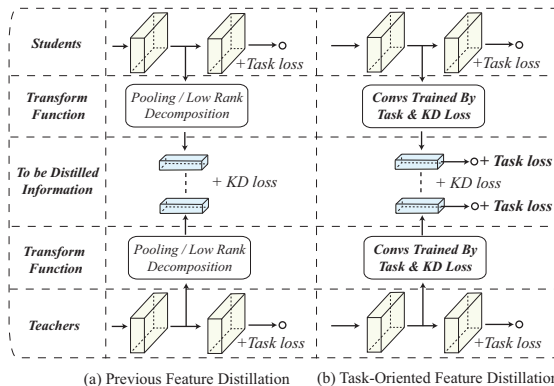

(a) Previous Feature Distillation     (b) Task-Oriented Feature Distillation

Figure 1: Comparison between previous feature distillation and task-oriented feature distillation.

---

[*]Equal contribution

[†]Corresponding authors

[3]https://github.com/ArchipLab-LinfengZhang/Task-Oriented-Feature-Distillation

Table 1: A survey of previous feature distillation methods and task-oriented feature distillation.

| Method | Transformation | Lost Information |
|---|---|---|
| AT [70] | Channelwise Pooling | Channel dims |
| FSP [68] | FSP Matrix | Spatial dims |
| Jacobian [58] | Gradients | Channel dims |
| SVD [26] | SVD Decomposition | Spatial dims |
| Heo *et al.* [20] | Margin ReLU | Negative Feature |
| Task-Oriented Feature Distillation | Convolutional Layers Trained by Task Loss | Non-Task-Oriented features |

into an easy-to-distill form and then distill them to students. In this progress, some unimportant information are filtered, as shown in Table 1. However, what still remains unknown is which form of information is the best to distill and which kind of transformation function can extract this form of information.

In this paper, *we assume that the task-oriented information is the information which is the most essential to distillation.* Based on this assumption, we propose a novel knowledge distillation method named task-oriented feature distillation (short as TOFD). Different from previous feature distillation methods whose transformation functions are manually designed, the transformation function in TOFD is convolutional layers which are trained in a data-driven manner by both distillation loss and the task loss. In the training period of TOFD, several auxiliary classifiers are attached at different depths to the backbone layers. Each auxiliary classifier consists of several convolutional layers, a pooling layer and a fully connected layer. They are trained to perform the same task as the whole neural network does. As a result, the auxiliary classifiers help to capture the task-oriented information from the whole features in the backbone layers, leading to high-efficiency knowledge distillation.

In most situations of knowledge distillation, the features of students and teachers have different widths, heights and channels. Usually, a convolutional layer or a fully connected layer is applied to match their sizes. However, this leads to one problem that some useful information of teachers may be lost in the progress of feature resizing. To address this problem, an orthogonal loss has been introduced in TOFD to regularize the weights of the feature resizing layer. With the property of orthogonality, more supervision from teachers can be exploited in students training.

Sufficient experiments demonstrate that the proposed TOFD achieves consistent and significant accuracy boost in various neural networks and datasets. Experiments in ten kinds of neural networks on five datasets show that TOFD outperforms the state-of-the-art distillation method by a large margin on both images classification and 3D classification. On average, 5.46%, 1.71%, 1.18%, 1.25% and 0.82% accuracy boost can be observed on CIFAR100, CIFAR10, ImageNet, ModelNet10, ModelNet40 datasets, respectively. Besides, ablation study and hyper-parameters sensitivity study are also conducted to show the effectiveness and stability of TOFD.

To sum up, the contribution of this paper can be summarized as follows:

- A novel knowledge distillation method named TOFD is proposed to distill the task-oriented information from teachers to students. Auxiliary classifiers are utilized to capture the task-oriented information from all features of teachers and students.

- An orthogonal loss is proposed to avoid the information loss of teacher's supervision in the feature resizing layers.

- Sufficient experiments on ten neural networks and five datasets are conducted to show the effectiveness of TOFD. Five kinds of knowledge distillation methods are utilized as the comparison experiments.

## 2 Related Work

### 2.1 Knowledge Distillation

Knowledge distillation is one of the most effective methods for model compression and acceleration. Bucila *et al.* first propose the idea of employing the large ensemble models to train a neural

network [6]. Hinton *et al.* further propose the concept of knowledge distillation and introduce a hyper-parameter "temperature" to control the training of student models. FitNet is proposed to train the student models with the feature map of teacher models instead of the logit[53]. Zagoruyko *et al.* apply feature distillation on the attention map of neural networks at different layers [70]. Furlanello *et al* sequentially train multiple student models and finally ensemble these models to achieve higher accuracy [14]. Shen *et al.* apply adversarial learning to knowledge distillation to minimize the difference between features of students and teachers [57]. Zhang *et al.* propose self-distillation, which distills the knowledge from deep layers to the shallow layers [71]. Besides image classification, knowledge distillation has also been applied in other domains, such as object detection [63], image segmentation [37], nature language processing [8, 59, 28], distributed training [2], semi-supervised learning [32]. Recently, more and more attention has been paid to study how to choose the best teacher model for distillation. Seyed-Iman *et al.* find that the teacher model with the highest accuracy is not the best teacher for distillation and the overlarge gap between the accuracy of students and teachers may harm the efficiency of knowledge distillation. They propose TAKD, which trains multiple teacher assistant models to facilitate knowledge distillation [45]. Based on the same observation, Jin *et al.* propose RKD, where the student models in different epochs are taught by teacher models in different epoches [29]. Cho *et al.* find that the teacher models trained with early-stopping always lead to accuracy boost on student models [10]. Moreover, Kang *et al.* and Liu *et al.* have applied neural network searching to find the student models for a given teacher model [31, 38].

## 2.2 Orthogonal Loss

As a desirable property in both convolutional neural networks and recurrent neural networks, the orthogonality of weights can solve the vanishing and exploding gradients problem by stabilizing the norm of gradients [61, 4, 19]. Therefore, sufficient methods have been proposed to obtain orthogonality in neural networks. Le *et al.* and Poole *et al.* apply the orthogonal loss to the autoencoder to encourage the neural networks to learn orthogonal representations of the inputs [47, 34]. Domain separation networks introduce the orthogonal loss to enable the shared and the private encoders to learn different aspects of the inputs [5]. Mhammedi *et al.* improve the efficiency of training the RNN model while ensuring its orthogonality through a new parameterization of the transition matrix [44]. Jing *et al.* propose the gated orthogonal recurrent units, which has the ability of capturing long term dependencies by orthogonal matrices [30]. Qi *et al.* apply the orthogonal loss to the transition matrix in PointNet to avoid the information loss [48]. Harandi *et al.* propose the generalized backpropagation which can be utilized to train neural layers with orthogonal weights [16]. Bansal *et al.* develop novel orthogonality regularization on the training of convolutional neural networks with mutual coherence and restricted isometry property [4]. Chernodub *et al.* propose the orthogonal permutation layer as a novel activation function to perform non-linear mappling [9].

## 3 Methodology

The details of the proposed task-oriented feature distillation are shown in Figure 2. It is observed that several auxiliary classifiers are attached at different depths of the convolutional neural networks. Each auxiliary classifier is composed of several convolutional layers, a pooling layer and a fully connected layer. They are trained to perform the same task as the whole neural network does. As a result, the convolutional layers in the auxiliary classifiers can capture the task-oriented information from the whole features. Then, these task-oriented information is distilled to the student models by $L_2$ loss. Moreover, to facilitate the training of auxiliary classifiers, a logit distillation loss is also applied to each pair of auxiliary classifiers between teacher models and student models. Note that these auxiliary classifiers are only utilized in the training period for knowledge distillation. They are not involved in the inference period, so there is no additional computation and parameters.

Another crucial problem is how to decide the number and the exact position of the auxiliary classifier. Recent progress in object detection and segmentation [52, 40] demonstrates that features with different resolutions have different information - low resolution features contain more information of the large objects while high resolution features contain more information of the small objects. Inspired by the above conclusion, we choose to perform TOFD before each downsampling layer in neural networks. As a result, different auxiliary classifiers can distill features of teacher models with different resolution. Note that the number of auxiliary classifiers is decided by the number of downsampling layers in the neural networks.

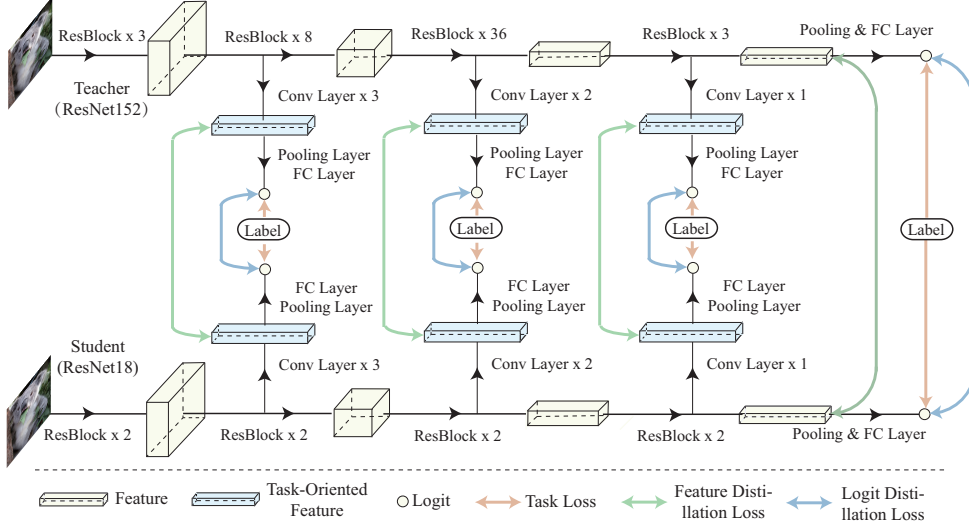

Figure 2: The overview of TOFD. Best viewed in color. (i) Several auxiliary classifiers are attached at different depths of the neural network. Each auxiliary classifier is composed of several conventional layers, a pooling layer and a fully connected layer. (ii) Each auxiliary classifier is trained by task loss so that it can capture the task-oriented information from the features. (iii) Feature distillation loss is applied between the task-oriented information of students and teachers. (iv) Logit distillation loss is also introduced to facilitate the training of auxiliary classifiers. (v) Auxiliary classifiers are dropped in the inference period to avoid additional computation and parameters.

## 3.1 Formulation

Let $\mathcal{X} = \{x_i\}_{i=1}^m$ be a set of training images, $\mathcal{Y} = \{y_i\}_{i=1}^m$ be the corresponding labels. Denote $F_i(\cdot)$ to be the feature map of the $i_{th}$ convolution block and $c_i(\cdot)$ to be the fully connected classifier in the $i_{th}$ convolution block. The superscript $t$ and $s$ denote the teacher model and student model respectively. In a neural network with $N$ convolutional blocks, the logit distillation [22] loss can be formulated as

$$\frac{1}{m}\sum_{i=1}^m \cdot L_{KL}(c_N^s(F_N^s(x_i)), c_N^t(F_N^t(x_i))), \tag{1}$$

where $L_{KL}$ is the KL divergence loss. The loss function of feature distillation can be formulated as

$$\frac{1}{m}\sum_{i=1}^m\sum_{j=1}^N L_2(T_j(F_j^s(x_i)), T_j(F_j^t(x_i))), \tag{2}$$

where $L_2$ is the $L_2$-norm loss and $T$ indicates the transformation function on the features. In most previous feature distillation methods, $T$ is a non-parametric transformation such as pooling and low rank decomposition. In contrast, $T$ in the proposed TOFD is several convolutional layers whose parameters are trained by both the task loss and the distillation loss. The proposed task-oriented feature distillation loss can be formulated as

$$\frac{1}{m}\sum_{i=1}^m\left\{\sum_{j=1}^N \alpha \cdot \underbrace{L_2(T_j(F_j^s(x_i)), T_j(F_j^t(x_i)))}_{\mathcal{L}_{\text{feature}}} + \underbrace{L_{CE}(c_j(T_j(F_N^s(x_i))), y_i)}_{\mathcal{L}_{\text{task}}}\right\}, \tag{3}$$

where $\alpha$ is a hyper-parameter to balance the two kinds of loss. Besides, we could further introduce the logit distillation loss to facilitate the training of the conventional transformation function $T$ and the fully connected layer $c$, which can be formulated as

$$\frac{1}{m}\sum_{i=1}^m\sum_{j=1}^N \underbrace{L_{KL}(c_j^s(T_j^s(F_N^s(x_i))), c_j^t(T_j^t(F_N^t(x_i))))}_{\mathcal{L}_{\text{logit}}}. \tag{4}$$

Table 2: Experiment results on CIFAR100 (Top-1 Accuracy /%). Numbers in bold are the highest.

| Model | Baseline | KD [22] | FitNet [53] | DML [73] | SD [71] | TOFD |
|---|---|---|---|---|---|---|
| ResNet18 | 77.09 | 78.34 | 78.57 | 78.72 | 78.64 | **82.92** |
| ResNet50 | 77.42 | 78.58 | 78.62 | 79.18 | 80.56 | **84.74** |
| PreactResNet18 | 76.05 | 77.41 | 78.79 | 77.03 | 78.12 | **82.06** |
| PreactResNet50 | 77.74 | 78.26 | 79.12 | 78.48 | 80.12 | **83.33** |
| SEResNet18 | 77.27 | 78.43 | 78.49 | 78.58 | 79.01 | **83.06** |
| SEResNet50 | 77.69 | 78.89 | 78.82 | 79.72 | 80.56 | **84.44** |
| ResNeXt50-4 | 79.49 | 80.46 | 79.54 | 80.39 | 82.45 | **84.67** |
| MobileNetV1 | 67.82 | 67.55 | 71.78 | 67.73 | 71.39 | **72.82** |
| MobileNetV2 | 69.04 | 70.16 | 70.21 | 68.79 | 71.45 | **73.57** |
| ShuffleNetV1 | 72.26 | 73.54 | 72.78 | 72.72 | 74.30 | **76.04** |
| ShuffleNetV2 | 72.38 | 72.86 | 74.36 | 72.66 | 73.32 | **76.68** |

## 3.2 Orthogonal Loss

In most distillation situations, the features of teachers and students have different sizes so their distance can not be directly minimized. To solve this problem, a convolutional or fully connected layer is always introduced to adjust their sizes. However, the information of teachers' features may be lost in this progress, which reduces the effectiveness of feature distillation. In this paper, we apply an orthogonal loss to the weights of the feature resizing layer to alleviate this problem. Denotes the distilled features of teacher models as the vector $\mathbf{x}$ and the weights of feature resizing layer as $\mathbf{W}$, the resized feature can be written as $\mathbf{Wx}$. To keep the feature information during the feature resizing process and inspired by Bansal *et al.* [4], we introduce an orthogonal loss that simultaneously penalizes the orthogonality of the row space and column space spanned by $\mathbf{W}$ in the feature resizing layer, i.e. the loss is defined as

$$\beta \cdot \underbrace{(\|\mathbf{W^T W} - \mathbf{I}\| + \|\mathbf{W W^T} - \mathbf{I}\|)}_{\mathcal{L}_{\text{orthogonal}}}, \tag{5}$$

where $\beta$ is a hyper-parameter to balance its magnitude and other loss. If a convolutional layer instead of a fully connected layer is utilized as the feature resizing layer, its weights can be first reshaped from $S \times H \times C \times M$ to $SHC \times M$ where S, H, C, M are width, height, input channel number and output channel number, respectively. To summarize, the overall loss function can be formulated as

$$\mathcal{L}_{overall} = \mathcal{L}_{\text{feature}} + \mathcal{L}_{\text{logit}} + \mathcal{L}_{\text{task}} + \mathcal{L}_{\text{orthogonal}}. \tag{6}$$

The overall loss function includes the feature distillation loss, logit distillation loss, task loss, orthogonal loss and two hyper-parameters. Ablation study and sensitivity study are introduced in Section 6 to demonstrate their effectiveness and sensitivity.

## 4 Experiment

### 4.1 Experiment Setting

**Image Classification.** The experiments of image classification are conducted with nine kinds of convolutional neural networks, including ResNet [17], PreActResNet [18], SENet [25], ResNeXt [65], MobileNetV1 [24], MobileNetV2 [55], ShuffleNetV1 [42], ShuffleNetV2 [43], WideResNet [69] and three datasets, including CIFAR100 and CIFAR10 [33], ImageNet [12]. In CIFAR experiment, each model is trained with 300 epochs by SGD optimizer and the batch size is 128. In ImageNet experiments, each model is trained with 90 epochs by SGD optimizer and the batch size is 256.

**3D Classification.** The experiments of point cloud classification are conducted with ResGCN [36] of different depths on two datasets including ModelNet10 and ModelNet40 [64]. Each model is trained with 100 epochs by Adam with learning rate decay in every 20 epochs.

**Comparison Experiments.** Four kinds of knowledge distillation methods have been utilized for comparison, including KD [22], FitNet [53], DML [73] and self-distillation [71]. All these experiments are repoduced by ourselves.

Table 3: Experiment results on CIFAR10 (Top-1 Accuracy /%). Numbers in bold are the highest.

| Model | Baseline | KD [22] | FitNet [53] | DML [73] | SD [71] | TOFD |
|---|---|---|---|---|---|---|
| ResNet18 | 94.25 | 94.67 | 95.57 | 95.19 | 95.87 | **96.92** |
| ResNet50 | 94.69 | 94.56 | 95.83 | 95.73 | 96.01 | **96.84** |
| PreactResNet18 | 94.20 | 93.74 | 95.22 | 94.80 | 95.08 | **96.49** |
| PreactResNet50 | 94.39 | 93.53 | 94.98 | 95.87 | 95.82 | **96.93** |
| SEResNet18 | 94.78 | 94.53 | 95.64 | 95.37 | 95.51 | **96.80** |
| SEResNet50 | 94.83 | 94.80 | 95.31 | 94.83 | 95.45 | **97.02** |
| ResNeXt50-4 | 94.49 | 95.41 | 95.78 | 95.41 | 96.01 | **97.09** |
| MobileNetV1 | 90.16 | 91.70 | 90.53 | 91.65 | 91.98 | **93.93** |
| MobileNetV2 | 90.43 | 92.86 | 90.49 | 90.49 | 91.02 | **93.34** |
| ShuffleNetV1 | 91.33 | 92.57 | 92.23 | 91.40 | 92.47 | **92.73** |
| ShuffleNetV2 | 90.88 | 92.42 | 91.83 | 91.87 | 92.51 | **93.74** |

Table 4: Experiment results on ImageNet (Top-1 Accuracy /%).

| Model | Baseline | TOFD | MAC(G) | Param(M) |
|---|---|---|---|---|
| ResNet18 | 69.76 | 70.92 | 1.82 | 11.69 |
| ResNet50 | 76.13 | 77.52 | 4.11 | 25.56 |
| ResNet101 | 77.37 | 78.64 | 7.83 | 44.55 |
| ResNet152 | 78.31 | 79.21 | 11.56 | 60.19 |
| ResNeXt50-32-4 | 77.62 | 78.93 | 4.26 | 25.03 |
| WideResNet50-2 | 78.47 | 79.52 | 11.43 | 68.88 |

## 4.2 Results on CIFAR10 and CIFAR100

Table 2 and 3 show the accuracy of student networks on CIFAR100 and CIFAR10. It is observed that: **(a)** The proposed TOFD leads to significant accuracy boost compared with the baseline models. In CIFAR100, 5.46% accuracy boost can be found on the eleven models on average, ranging from 6.75% at SENet50 as the maximum to 3.78% at ShuffleNetV1 as the minimum. In CIFAR10, 2.49% accuracy boost can be found on the eleven models on average, ranging from 3.77% at MobileNetV1 as the maximum to 1.40% at ShuffleNetV1 as the minimum. **(b)** In all the models, the proposed TOFD outperforms the second best distillation method by a large margin. On average, 3.13% and 1.28% accuracy boost compared with the second best distillation method can be observed on CIFAR100 and CIFAR10, respectively. **(c)** The proposed TOFD not only works on the over-parameters models such as ResNet and SENet, but also shows significant effectiveness in the lightweight models such as MobileNet and ShuffleNet. On average, 4.40% and 2.74% accuracy boost of the lightweight models can be observed on CIFAR100 and CIFAR10 datasets.

## 4.3 Results on ImageNet

Table 4 shows the experiment results of TOFD on ImageNet. ResNet152 model is utilized as the teacher model across all these experiments. It is observed that **(a)** On average, TOFD leads to 1.18% accuracy improvements across the 6 neural networks. **(b)** The distilled ResNet50 and ResNet101 have higher accuracy than the baselines of ResNet101 and ResNet152 respectively. By replacing the distilled ResNet50 and ResNet101 with ResNet101 and ResNet152 respectively, TOFD achieves 1.57 times compression and 1.81 acceleration with no accuracy loss.

## 4.4 Results on ModelNet10 and ModelNet40

Table 5 and  6 show the experiment results of TOFD on ModelNet10 and ModelNet40. It is observed that **(a)** In 3D classification tasks, knowledge distillation methods are not as effective as they do in image classification tasks. In the five distillation methods, only DML and TOFD can achieve consistent accuracy boost than the baseline. **(b)** TOFD outperforms other knowledge distillation methods on all models and datasets. Compared with the baselines, 1.25% and 0.82% accuracy boost can be found in ModelNet10 and ModelNet40 with TOFD on average.

Table 5: Experiments results of the 3D classification task on ModelNet10 (Top-1 Accuracy /%). Numbers in bold are the highest.

| Model | Baseline | KD [22] | FitNet [53] | DML [73] | SD [71] | TOFD |
|---|---|---|---|---|---|---|
| ResGCN8 | 92.73 | 93.50 | 94.05 | 93.61 | 93.17 | **94.38** |
| ResGCN12 | 93.50 | 93.28 | 93.17 | 94.05 | 92.40 | **94.16** |
| ResGCN16 | 92.40 | 92.84 | 93.39 | 93.17 | 92.62 | **93.83** |

Table 6: Experiments results of the 3D classification task on ModelNet40 (Top-1 Accuracy /%). Numbers in bold are the highest.

| Model | Baseline | KD [22] | FitNet [53] | DML [73] | SD [71] | TOFD |
|---|---|---|---|---|---|---|
| ResGCN8 | 90.76 | 91.29 | 90.76 | 91.69 | 90.60 | **91.77** |
| ResGCN12 | 90.32 | 91.21 | 90.80 | 91.41 | 90.80 | **91.65** |
| ResGCN16 | 91.33 | 91.45 | **91.45** | 91.33 | 91.25 | **91.45** |

## 5  Discussion

### 5.1  Do Auxiliary Classifiers Really Capture Task-Oriented Information?

The auxiliary classifiers in TOFD are introduced to capture the task-oriented information from the features of both students and teachers. As shown in Figure 3, we have visualized the features in the backbone layers and the task-oriented features captured by the auxiliary classifiers with the Gram-Cam method [56]. It is observed that: **(a)** Except the features of the last layer (sub-figure d), the features in the backbone layers have no direct relation with the classification task. The attention of convolutional layers are paid to the whole figure uniformly, indicating there is much non-task-oriented information in the features of backbone layers. **(b)** In the heatmaps of the auxiliary classifier, the pixels of the dog have much more attention value than the background, indicating that auxiliary classifiers really capture the task-oriented information from the original features.

### 5.2  Ablation Study

As shown in Table 7, an ablation study on CIFAR100 with ResNet18 has been conducted to demonstrate the individual effectiveness of different components in TOFD. It is observed that **(a)** Compared with the combination between feature distillation and logit distillation, 3.50% (82.31%-78.81%) accuracy boost can be obtained with the auxiliary classifiers, indicating that the task-oriented information is beneficial to knowledge distillation. **(b)** With only the auxiliary classifier, 2.90% (77.09%-79.99%) accuracy boost can be observed compared with the baseline, indicating that the multi-exit training itself can facilitate model training. **(c)** The orthogonal loss on feature resizing layer leads to 0.61% (82.92%-82.31%) accuracy boost.

### 5.3  Sensitivity Study on Hyper-parameters $\alpha$ and $\beta$

The hyper-parameters $\alpha$ and $\beta$ are introduced in TOFD to control the magnitude of feature distillation loss and the orthogonal loss. As shown in Figure 4 and Figure 5, experiments on CIFAR100 and ResNet18 have been conducted to study their sensitivity. It is observed that: **(a)** Even in the worst situation when $\alpha = 0.01$, TOFD still achieves 5.48% accuracy improvements than the baseline and

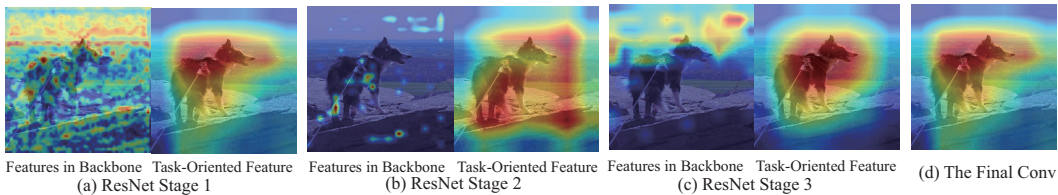

Features in Backbone   Task-Oriented Feature   Features in Backbone   Task-Oriented Feature   Features in Backbone   Task-Oriented Feature   (d) The Final Conv
(a) ResNet Stage 1                              (b) ResNet Stage 2                              (c) ResNet Stage 3

Figure 3: Comparison on the Grad-CAM [56] visualization results between the features of the backbone layers and the task-oriented features captured by auxiliary classifiers.

Table 7: Ablation study with ResNet18 on CIFAR100 (Top-1 Accuracy /%).

| $\mathcal{L}_{\text{logit}}$ | × | ✓ | × | ✓ | × | ✓ | ✓ |
|---|---|---|---|---|---|---|---|
| $\mathcal{L}_{\text{feature}}$ | × | × | ✓ | ✓ | × | ✓ | ✓ |
| $\mathcal{L}_{\text{task}}$ | × | × | × | × | ✓ | ✓ | ✓ |
| $\mathcal{L}_{\text{orthogonal}}$ | × | × | × | × | × | × | ✓ |
| Accuracy | 77.09 | 78.34 | 78.57 | 78.81 | 79.99 | 82.31 | 82.92 |

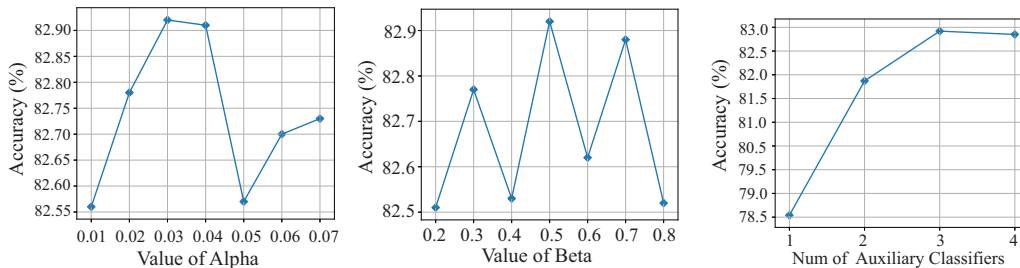

Figure 4: Sensitivity study on $\alpha$. Figure 5: Sensitivity study on $\beta$. Figure 6: TOFD with different numbers of auxiliary classifiers.

3.93% accuracy improvement than the second best knowledge distillation method (SD [71]). **(b)** When $\beta$ ranges from 0.2 to 0.8, the accuracy of TOFD ranges from 82.51% to 82.92%. Even in the worst situation when $\beta = 0.2$, TOFD still achieves 5.42% accuracy improvements than the baseline and 3.87% accuracy improvement than the second best knowledge distillation method (SD [71]). These experiment results demonstrate that TOFD is not sensitive to the value of hyper-parameters.

### 5.4 How does the Number of Auxiliary Classifiers Influence TOFD?

In TOFD, several auxiliary classifiers are introduced in the training period to capture the task-oriented information from all the features. In this section, a series of experiments are conducted on CIFAR100 with ResNet18 to show how the number of auxiliary classifiers in TOFD influences the model accuracy. As shown in Figure 6, when there are less than 4 auxiliary classifiers in the neural network, each auxiliary classifier leads to a significant accuracy boost. In contrast, not only can the fourth auxiliary classifier hardly improve model accuracy, it even leads to a small amount of accuracy drop which indicates that too many auxiliary classifiers may result in the over-regularization problem.

## 6 Conclusion

In this paper, we propose a novel knowledge distillation method named task-oriented feature distillation (TOFD). Based on the assumption that the task-oriented feature in neural networks is more essential for distillation, we attach several auxiliary classifiers at different depths of models. Since the auxiliary classifiers are trained to perform the same task as the whole neural network does, they are able to capture the task-oriented information of teacher models. As a result, TOFD enables teachers to distill only the task-oriented information to the students, which leads to significant and consistent accuracy improvements on various neural networks and datasets. Besides, we have applied an orthogonal loss to the feature resizing layer in TOFD to further boost the performance of student models. Experiments on 10 kinds of neural networks and 5 datasets demonstrate that TOFD has consistent and significant effectiveness. Abundant ablation study and hyper-parameters sensitivity study are also conducted to demonstrate the stability of TOFD. The Grad-CAM visualization results show that auxiliary classifies do capture the task-oriented information.

The excellent performance of TOFD indicates that the task-oriented features are more essential in the knowledge transfer from teachers to students. This idea may be extended to the other domains in deep learning, such as domain adaptation.

# 7 Appendix

In this section, we show the additional comparison experiments which are required by reviewers. Table 8 shows the comparison experiments with the other 6 kinds of knowledge distillation methods. It is observed that our method outperforms the second-best knowledge distillation methods by 3.71%, 5.02%, 2.55% on ResNet18, ResNet50 and MobileNetV2 on CIFAR100, respectively. Table 9 shows the comparison experiments on ImageNet. It is observed that our method leads to 1.16%, 1.29% and 1.38% accuracy improvements on ResNet18, MobileNetV2 and ShuffleNetV2, respectively.

Table 8: Comparison with more KD methods on CIFAR100.

| Model | VID [1] | AT [70] | FSP [68] | Jacob [58] | SVD [26] | Heo [21] | Ours |
|---|---|---|---|---|---|---|---|
| ResNet18 | 78.93 | 78.45 | 78.75 | 78.45 | 78.53 | 79.21 | **82.92** |
| ResNet50 | 79.21 | 78.73 | 79.02 | 79.03 | 78.82 | 79.72 | **84.74** |
| MobileNetV2 | 70.62 | 70.34 | 70.48 | 70.26 | 69.35 | 71.02 | **73.57** |

Table 9: Comparison with other KD methods on ImageNet.

| Model | Baseline | KD [22] | FitNet [53] | SD [71] | DML [73] | Ours |
|---|---|---|---|---|---|---|
| ResNet18 | 69.76 | 70.45 | 70.26 | 70.51 | 70.39 | **70.92** |
| MobileNetV2 | 71.52 | 72.23 | 71.95 | 72.37 | 72.29 | **72.81** |
| ShuffleNetV2 | 69.36 | 70.14 | 69.86 | 70.26 | 70.21 | **70.74** |

# 8 Acknowledgements and Funding Disclosure

This work was partially supported by IIISCT (Institute for interdisciplinary Information Core Technology), National Natural Sciences Foundation of China (No.31970972 and 11901338), and Tsinghua University Initiative Scientific Research Program. Besides, we sincerely appreciate all reviewers for their detailed and thoughtful reviews.

# 9 Broader Impact

The proposed TOFD is a novel knowledge distillation method, which can be utilized in the training of all kinds of neural networks. Since TOFD is not designed for a specific application, it's impact on the society may not be very obvious. Its potential impact can be summarized as follows.

**Positive.** TOFD can be utilized to compress and accelerate the neural networks or improve their accuracy. As a result, it can facilitate the computer vision application in resource-limited edge devices,

such as mobile phones, self-driving cars and embedding devices and so on. Moreover, by reducing the size of neural networks, TOFD can reduce the energy consumption of neural networks, making them more environment-friendly.

**Negative.** Unfortunately, computer vision techniques, if used improperly, or without permission, may have the potential for violation of image rights. This should be regularized by law.

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
