[Supplementary Material · Supplementary_Material.pdf]

# Task-Oriented Feature Distillation
# Supplementary Material

## 1 Architecture of Auxiliary Classifiers

The auxiliary classifier is composed of several depthwise separable convolution layers, a pooling layer, and a fully connected layer. Depthwise separable convolution is proposed in MobileNet [5] to reduce the size of the vanilla convolutional layer. To reduce the training overhead of TOFD, we apply depthwise separable convolution in the auxiliary classifiers instead of the vanilla convolutional layers. Take ResNet [4] as an example, the first, second, third auxiliary classifier has 3, 2, 1 depthwise separable convolution layers, respectively. Besides, a fully connected and a global average pooling layer are attached after the depthwise separable convolutional layers. Note that the stride of each depthwise separable convolution is 2, the kernel size is 3, and the padding is 1.

Table 1: Sensitivity study of the choice of teachers.

| Teacher Model | Student Accuracy |
|---------------|------------------|
| ResNet18      | 82.65            |
| ResNet50      | 82.45            |
| ResNet101     | 82.92            |
| ResNet152     | 82.47            |

## 2 Influence from Teachers

Recent researchers in knowledge distillation show that the performance of students is sensitive to the selection of the teacher model and a teacher model with the highest accuracy may not be the best teacher model for knowledge distillation. [6, 2]. In this section, we show how does the performance of student models influenced by the choice of teachers. As shown in Table 1, we show the accuracy (%) of a ResNet18 student model trained on CIFAR100 by the proposed TOFD with different teacher models. It is observed that the range between the best teacher (resnet101) and the worst teacher (resnet50) is less than 0.50%, indicating that TOFD is not very sensitive to the choice of teacher models. In the image classification task, we usually take ResNet101 and ResNet152 as the teacher. In the 3D classification task, we take the ResGCN16 model as the teacher.

## 3 More Examples of Figure 3

In Figure.3 of the submitted paper, we have compared the Grad-CAM [7] visualization results of features in the backbone and the task-oriented features captured by the convolutional layers in the auxiliary classifier. Due to the limitation of paper length, we only put one example in the paper. Here we show the visualization results of another four images.

Figure 1: Comparison on the Grad-CAM [7] visualization results between the features in the backbone layers and the task-oriented features captured by the convolutional layers in the auxiliary classifiers.

## 4 Implementation Tricks

**Cut-Out Data Augmentation.** Besides the basic data augmentation methods such as padding, cropping, horizon flipping, we have applied the cut-out [3] data augmentation in the experiments on image classification tasks.

**Orthogonal Loss Decay.** As discussed by Bansal *et al.* [1], the orthogonal loss may have a negative impact at the end of the training period. We take their regularization coefficients decay scheme to alleviate this issue.