[Reviews · NeurIPS 2020]

Review 1

Summary and Contributions: The authors propose Task-Oriented Feature Distillation (TOFD). The technique augments a normal teacher-student training with auxiliary networks to extract features that are critical for predictions for the given task. In contrast, most existing KD methods simply use some manual transformations of the intermediate layers from teachers and students to get KD losses. Several experiments are presented with datasets ranging from CIFAR-10/100, Imagenet to ModelNet.

Strengths: The paper has following strengths: 1. Idea is good and makes intuitive sense. Transferring task-specific information to the students should help them learn better. 2. Accuracy improvement is clear (although there are some concerns with lack of comparisons and missing experiments). 3. A lot of ablation studies are conducted which clearly explain how much improvement can be attributed to which part of the approach. 4. The paper is well-written (with some minor typos).

Weaknesses: Despite the strengths, the reviewer identified the following weaknesses: 1. No comparisons against baseline KD methods for Imagenet. 2. Can the authors show distillation results for Mobilenet-like compact baselines for Imagenet? 3. DML and TOFD are too close for 3D classification experiments. What was the standard dev. and how many times were each of the experiments repeated? Also, since they are close (and because the orthogonal loss is basically a prior work), the authors should have augmented the baselines with the orthogonal loss as well for a fairer comparison. For 3D classification, it is possible that DML will outperform TOFD when augmented with orthogonal loss. 4. The authors could have used a few other recent baselines for comparison (e.g., https://openaccess.thecvf.com/content_CVPR_2019/papers/Ahn_Variational_Information_Distillation_for_Knowledge_Transfer_CVPR_2019_paper.pdf) 5. Why were hyperparameters for \cal{L}_{task} and \cal{L}_{logits} fixed to 1 and not varied? 6. Lack of theoretical insights. Example: Is there a way to find out how to select how many auxiliary networks we need? 7. Please remove the claim that "the authors are the first to use orthogonal loss for KD". It does not sound like a big contribution as it is directly used from prior works. The first two points above are the most critical for this reviewer. If these are addressed, the reviewer will consider updating the rating. ========== UPDATE AFTER REBUTTAL =========== I have increased my rating from 5 to 6 since the authors have addressed most of my concerns in the rebuttal. The reason I am not increasing the rating more than 6 is because the paper still lacks theoretical insights but offers good empirical results overall.

Correctness: Yes.

Clarity: Yes.

Relation to Prior Work: Yes.

Reproducibility: Yes

Additional Feedback: Please see above.


Review 2

Summary and Contributions: The paper proposes a modified distillation method, called task-oriented feature distillation (TOFD) which uses auxiliary losses (and classifiers) during distillation. The paper provides empirical evidence for the effectiveness of TOFD on image classification and 3D classification tasks. Edit: Revised score I am glad that the paper looks much stronger now (with the additional results provided by the authors). I am not satisfied with the response about the case where the backbone network fails to learn useful features. While the authors response is technically correct, it is not easy to calibrate the capacity of the additional models or the right choice of \alpha, \beta. Hence I am not sure how to design these additional models. Overall I am happy with the author's response and I am increasing my score from 5 to 6.

Strengths: * The proposed method, while complicates the training setup, does not complicate the evaluation setup. * Good number of experiments - 10 model architectures and five datasets (across two domains). * Ablation study to understand the contribution of different components of the proposed setup. Interestingly, combining feature + logit distillation makes the most significant difference. * While the paper may appear to have limited novelty, I think it is useful to show that the different existing ideas can be combined (without negative interference).

Weaknesses: * Line 141, the paper mentions that the auxiliary classifiers are trained with both task loss and distillation loss. While I understand the motivation behind using the task loss, why should these classifiers minimize the distillation loss? How do we avoid the case where the classifiers have enough capacity to minimize the distillation loss and the supervised learning loss, and the backbone model does not have to learn useful features. Would this not defeat the purpose of adding these classifiers. * Effect of the extra capacity. Adding all the additional classifiers increases the capacity of the model. For a fair comparison, the capacity of the baseline models should also be increased. I think the argument that "extra parameters are only used during training" is insufficient as we do not know the effect of overparameterization during distillation. To be clear, I am not asking that the extra classifiers be added to all the baselines. I think making the baselines wider/deeper should be sufficient. * The paper lists certain models (and techniques for transforming the features) in table 1, but does not use them as baselines for the empirical evaluation. This lowers my confidence in the proposed approach.

Correctness: Given the experimental setup, the claims are valid.

Clarity: Paper is largely well written and easy to read.

Relation to Prior Work: Yes

Reproducibility: Yes

Additional Feedback: Could the authors clarify the loss function for the baselines - Are they training the baselines with just the distillation loss or supervised loss + distillation loss. In practice, I have seen people use both.


Review 3

Summary and Contributions: The paper introduces a novel feature distillation approach that adds auxiliary classifiers and the corresponding task-oriented loss functions at intermediate network layers, to better guide feature distillation from teacher to student model. Extensive empirical results on a wide range of network architectures and datasets demonstrate consistent performance improvements as compared with several state-of-art distillation approaches.

Strengths: The authors propose a relatively straightforward but empirically beneficial extension to state-of-art feature distillation approaches, consisting in adding task-oriented objective functions to intermediate network layers and thus enforcing more task-relevant feature distillation; as an additional regularizer, orthogonal loss is added in feature-resizing layers. Empirical evaluation appears quite solid, on 10 different network architectures and 5 datasets, and with 5 baseline distillation methods, and shows consistent and noticeable improvements. The approach is novel, though the contribution seems to be somewhat straightforward and the results are not very surprising. Still, performance improvements are convincing and relevant/useful to the NeurIPS community.

Weaknesses: As noted above, the contribution is somewhat incremental and not very surprising, though it is reassuring to see empirical support in favor of the proposed idea that explicitly including task-related objectives in feature distillation at intermediate network levels leads to better predictive features.

Correctness: Empirical methodology is solid and extensive. (the paper does not make any theoretical claims).

Clarity: The paper is clearly written and easy to follow.

Relation to Prior Work: The discussion of prior work is quite comprehensive and clear.

Reproducibility: Yes

Additional Feedback:


Review 4

Summary and Contributions: - Originality: The paper is moderately original as the authors mention. Authors propose a assumption that task-oriented information is essential to distillation. And from experiment, proposed method achieve state-of-the-art performance. - Clarity: The paper is partly well written and part of my confusions has been resolved from ablation study. - Significance: This paper is likely to be of interest to the NeurIPS community. The authors report good results and method proposed is easy to implement to most neural networks.

Strengths: - Originality: The paper is moderately original as the authors mention. Authors propose an assumption that task-oriented information is essential to distillation. And from experiment, proposed method achieve state-of-the-art performance. - Clarity: The paper is partly well written and part of my confusions has been resolved from ablation study. - Significance: This paper is likely to be of interest to the NeurIPS community. The authors report good results and method proposed is easy to implement to most neural networks.

Weaknesses: 1) The authors propose a view that the orthogonal loss can ease the problem of information loss of teacher. However, the paper lacks insight into this view. Even though orthogonal loss can further improve the performance of the proposed method. It is not clear how orthogonal loss can prevent teacher information from being lost. 2) Task loss do improve the performance of proposed method form experiments. However, It can hardly infer that there exists strong relationship between knowledge distillation learning and task loss. From Figure 3, we can only infer that the task loss can help model capture the task-oriented information from the original features, but whether task-oriented information in the low-feature can help knowledge distillation learning is unclear. [Post-rebuttal Updates] The rebuttal is satisfied to me, except the insight of the proposed loss and the empirical study to support this insight. I change the final rating from 5 to 6.

Correctness: The claims and method are clear but empirical analysis is not convincing enough.

Clarity: Partially.

Relation to Prior Work: The paper is moderately original as the authors mention.

Reproducibility: Yes

Additional Feedback:

[Author Response · NeurIPS 2020]

Table 1: Comparison with other KD methods on CIFAR100.

| Model | Baseline | VID | AT | FSP | Jacobian | SVD | Heo | Ours |
|---|---|---|---|---|---|---|---|---|
| ResNet18 | 77.09 | 78.93 | 78.45 | 78.75 | 78.45 | 78.53 | 79.21 | **82.92** |
| ResNet50 | 77.42 | 79.21 | 78.73 | 79.02 | 79.03 | 78.82 | 79.72 | **84.74** |
| MobileNetV2 | 69.04 | 70.62 | 70.34 | 70.48 | 70.26 | 69.35 | 71.02 | **73.57** |

Table 2: Comparison experiments on ImageNet.

| Model | Baseline | KD | FitNet | Self-Distill | DML | Ours |
|---|---|---|---|---|---|---|
| ResNet18 | 69.76 | 70.45 | 70.26 | 70.51 | 70.39 | **70.92** |
| MobileNetV2 | 71.52 | 72.23 | 71.95 | 72.37 | 72.29 | **72.81** |
| ShuffleNetV2 | 69.36 | 70.14 | 69.86 | 70.26 | 70.21 | **70.74** |

We sincerely appreciate all reviewers for their most detailed and thoughtful reviews.

**Overall Response: Comparison with more KD methods (R1'Question-3 and R4'Question-2).** Tab.1 shows the
comparison with additional 6 kinds of KD methods, including the VID (Ahn *et al.*, CVPR2019) requested by R1, and
5 kinds of KD methods (from Tab.1 of the paper) requested by R4. It is observed that our method outperforms the
second-best KD method by 3.71%, 5.02%, 2.55% on ResNet18, ResNet50, MobileNetV2 @ CIFAR100, respectively.

**Response to R1: Question-1 and Question-2.** Tab.2 shows the comparison experiments on ImageNet with ResNet18,
MobileNetV2 and ShuffleNetV2. It is observed that our method leads to 1.16%, 1.29%, 1.38% accuracy improvements
on ImageNet with ResNet18, MobileNetV2, ShuffleNetv2 respectively, outperforming the second-best KD method
by 0.41%, 0.44%, 0.48%, respectively. **Question-3.** We have re-conducted experiments on ModelNet10/40 with
ResGCN8 and each experiment is repeated 5 times. On ModelNet10, the accuracy of our method and DML are
$94.35 \pm 0.21\%$ and $93.63 \pm 0.17\%$, respectively. On ModelNet40, the the accuracy of our method and DML are
$91.78\pm0.12\%$ and $91.67 \pm 0.09\%$, respectively. Note that DML is a logit-based KD method while the orthogonal loss
can only be utilized in feature-based KD methods so DML can not be improved by the orthogonal loss. **Question-4.**
Please refer to the overall response (the VID Column). **Question-5.** We do not introduce new hyper-parameters for
balancing these two losses because we find that the fixed 1:1 loss ratio is good enough. Additional experiments show
that the performance of our method can be improved by 0.15% by introducing and adjusting a new hyper-parameter
here (ResNet18, CIFAR100). **Question-6.** The proposed method has given strong experimental clue to the intuition
that *task-oriented features are more essential in the knowledge transfer related domains.* These improvements might
open up new research opportunities for exploring the transfer-based ML including mathematical theory and algorithm.
**Question-7.** Thanks for your advice, we will remove this sentence in the final version.

**Response to R4: Reply to additional feedback.** The training loss for KD baselines includes both distillation loss
and the supervised loss. Thanks for the valuable question and we will modify our writing to clarify this confusion.
**Question-1.** We apply both supervised loss and distillation loss to the auxiliary classifiers instead of only supervised
loss because the distillation loss can facilitate the training of auxiliary classifiers. A more accurate auxiliary classifier
is able to extract better task-oriented features, which improves the performance of knowledge distillation in turn. *How*
*to Avoid the Case?* The case that auxiliary classifiers have learned to minimize their loss but the backbone model fails
to learn useful features will not happen because (i) Compared with the whole backbone, the auxiliary classifiers have
much fewer parameters so they have weaker learning ability than the backbone model. (ii) The auxiliary classifiers
are trained to minimize both distillation loss and supervised loss, which is harder than only the supervised loss. (iii) In
the overall loss function, we multiply the loss of auxiliary classifiers by hyper-parameters whose value is less than 1
($\alpha = 0.03$, $\beta = 0.5$), which forces the whole model to pay more attention to the learning of the backbone and the final
classifier, instead of the auxiliary classifiers. **Question-2.** With the same or fewer parameters in the training period,
our method still outperforms the other KD methods. For instance, ResNet18 with our method has 12.10M and 11.22M
parameters in the training and testing period respectively, achieving 82.92% accuracy in the testing period. In contrast,
ResNet50 with Hinto's KD has 23.70M parameters in both the training and testing period, achieving 78.58% accuracy
in the testing period. Note that the parameters of auxiliary classifiers are very tiny (only 7% of the whole model, on
ResNet18). **Question-3.** We have added comparison experiments with them. Please refer to the overall response.

**Response to R5:** The proposed method has given strong experimental clue to the intuition that *task-oriented features*
*are more essential in the knowledge transfer related domains.* These improvements might open up new research
opportunities for exploring the transfer-based ML including mathematical theory and algorithm.

**Response to R6: Question-1.** This question can be considered in two aspects. First, when the feature resizing layer is
used to increases the dimension of student features, due to the energy-preserving property of orthogonality, the energy
of student features will not be amplified by the feature resizing layer. Second, when the feature resizing layer is used
to decrease the dimension of teacher features, the weight of feature resizing layer works as an orthogonal projection
to the teacher features, which leads to less information loss than non-orthogonal projection. **Question-2.** As shown in
the ablation study (Tab.7), compared with naive feature distillation, the task loss brings 3.50% accuracy improvement,
which accounts for 80% of the total accuracy gain, indicating that the task loss is the key of our method.

[Meta-Review · NeurIPS 2020]

The paper proposes task-oriented feature distillation, which introduces additional distillation objective with task loss from intermediate layers of the network. Although the idea is simple, it’s reasonable and well motivated. The experimental results show improved classification performance over the baselines. On the negative side, the novelty of the method seems incremental. With all these being said, the reviewers were mostly satisfied with the rebuttal and converged in favor of accepting the paper.